# The 17-Item Computer Vision Symptom Scale Questionnaire (CVSS17): Translation, Validation and Reliability of the Italian Version

**DOI:** 10.3390/ijerph19052517

**Published:** 2022-02-22

**Authors:** Gemma Caterina Maria Rossi, Federica Bettio, Mariano González-Pérez, Aba Briola, Gemma Ludovica Maria Pasinetti, Luigia Scudeller

**Affiliations:** 1Surgical Sciences Department, UOC Oculistica, University Eye Clinic, Fondazione IRCCS Policlinico S. Matteo, 27100 Pavia, Italy; f.bettio@gmail.com (F.B.); a.briola@smatteo.pv.it (A.B.); 2Faculty of Optics and Optometry, Universidad Complutense de Madrid, 28001 Madrid, Spain; mgonzalez@gmail.com; 3Liceo Ugo Foscolo, 27100 Pavia, Italy; gemma.pasinetti@gmail.com; 4Research and Innovation Unit, IRCCS Azienda Ospedaliero-Universitaria di Bologna, 40121 Bologna, Italy; luigia.scudeller@aosp.bo.it

**Keywords:** video display terminal workers, VDT, quality of life, ocular surface, asthenopia, computer vision symptom scale questionnaire, CVSS17, computer vision syndrome, dry eye, ocular surface disease, COVID-19

## Abstract

Background. To validate the 17-item Computer Vision Symptom Scale questionnaire (CVSS17) in Italian. Methods. Cross-sectional validation study on video terminal (VDT) users and a reference sample of subjects not working at a VDT (control group), cognitively able to respond to a health status interview. The Italian self-administered version of the CVSS17 questionnaire was administered to all participants. The reliability and validity of the Italian translation of the CVSS17 were tested using standard statistical methods for questionnaire validation. The Rasch analysis was performed as well. Results. A total of 216 subjects were enrolled. Concerning the reliability, the Cronbach’s alpha coefficient was 0.925 (from 0.917 to 0.924), and the test–retest stability was 0.91 (<0.001). Concerning the validity, the control group had significantly better scores, and there were good correlations between responses to the CVSS17 and analogous domains of the GSS. Conclusion. The Italian version of the CVSS17 has shown psychometric properties comparable to those of the Spanish version, having good validity, discriminatory power, internal consistency and reliability. The questionnaire is a specific measure of vision-related quality of life in Italian-speaking VDT workers and can be used both in clinical practice and for research purposes.

## 1. Introduction

Video display terminal (VDT) users often report computer-related visual and ocular symptoms, which are the most common health problems among these subjects.

Given the high prevalence of these concerns in VDT workers [1,2,3], the Italian law has introduced an eye examination to assess ocular status in VDT workers (Legislative Decree 81/08 ex legge 626/96).

Studies to estimate the prevalence of computer-related visual and ocular symptoms were based on specific questionnaires designed by some authors [4,5].

Recently, a Spanish group developed and validated the Computer Vision Symptom Scale (CVSS17) questionnaire to assess visual and ocular symptoms in VDT workers: it includes 17 items and has been designed and validated for the Spanish language and culture. The Spanish version measures are valid and reliable, and what it measures correlates with some clinical outcome [6,7,8].

Before using a questionnaire that has been validated in a particular language, it is necessary to follow some validation procedures: the questionnaire must be adequately translated into another language and the translated questionnaire must be revalidated to prove that it is equivalent to the original [9]. In fact, the translation of a questionnaire into another language can compromise its validity [10,11] if the questions and response categories do not have the same meaning for patients and for researchers [12], and also if the original version and the translated one do not match. Consequently, the psychometric properties of the questionnaire must be re-established within the new cultural and linguistic context to achieve a correct transcultural adaptation [13,14].

The aim of the present study was to validate the Italian version of the 17-item Computer Vision Symptom Scale questionnaire, to make a tool available for assessing the symptoms of ocular strain from the patient’s point of view and to be used for all subjects with ocular symptoms due to prolonged stays at a VDT.

## 2. Materials and Methods

The present study was a single-center, observational, cross-sectional validation study. It was carried out at the University Eye Clinic of Pavia according to the Declaration of Helsinki with the approval of the Ethics Committee of the Fondazione IRCCS Policlinico San Matteo, Pavia (Proc. N. 20140038305). All subjects gave written informed consent before enrollment.

The CVSS17 is the final version of the pilot version of 77 items (CVSS77). The questionnaire contains 17 items with different rating scales. Two items have two response categories, eleven items have three response categories and four items have four response categories. The questionnaire gives information about 15 different symptoms, considering a symptom’s severity and frequency, and the subject’s opinion. The high internal consistency (Cronbach’s α = 0.92) of the Spanish version makes it useful for comparison between groups and for clinical application, for both genders and both for presbyopes and for non-presbyopes.

The steps for getting the Italian adaptation process were: translation into Italian, assessment of item comprehension, backtranslation into English, development of a consensual version, and formal assessment of its validity and reliability [15]. Two bilingual translators (GLP and GCR) translated the questionnaire into Italian (GCMR and GMP), rating difficulty from 0 (lowest) to 100 (highest). A conceptual rather than literal equivalence was aimed at and discrepancies were reconciled. Another translator (LS) rated the reconciled forward translation for conceptual equivalence to the Spanish version (scale 1–100). The translated questionnaire was read to 10 Italian subjects to verify the clarity, unambiguity and completeness of each item. A bilingual Spanish-speaking translator (Isabel Mazzucchelli) translated back the questionnaire into English and then rated the equivalence to the original Spanish version (1–100). The Italian version was then discussed by a panel of experts to obtain content correspondence. We consulted the method for crosscultural equivalence in psychiatric research described by Ware for the IQOLA project in 1998 [16] and McAlinden in 2011 [17]. A pretest was performed on a sample composed of 35 subjects to analyze, review and correct issues related to the translation [14,18,19]: no question required to be corrected or revised.

Eligibility criteria were: age 18 years or older, no ocular surgery in the preceding 6 months, signed written consent, VDT workers as defined by the Italian law (>20 h a week on a PC); in the control group, subjects were not VDT workers.

The employees of the San Matteo Foundation who operate as VDT workers and who carried out an eye examination for occupational medicine, were invited to participate in the study for the VDT group; patients who underwent an outpatient eye examination and who were not VDT workers were invited to participate in the study for the control group.

For each subject, all the following variables were collected: age and gender, best corrected near and distance visual acuity (VA) and current refractive correction, years since working at a VDT, number of effective hours at a VDT per day, number of breaks per day and time spent on breaks, tear film break-up time (tf-BUT), corneal staining and Glaucoma Symptom Scale (GSS) questionnaire.

All the subjects self-completed the questionnaire with paper and pencil prior to the visit with the ophthalmologist to avoid interference with visit results (communication of bad/good news). To assess repeatability, a subset of 20 subjects was asked to self-complete the questionnaire twice: one on the appointed day of the visit and one 5–7 days later.

In order to provide data for the planned tests of correlational validity, the Italian version of the Glaucoma Symptom Scale (GSS) was administered in addition to the CVSS17. The GSS was chosen because it had been adequately translated and validated in Italian in 2013 [20]. The GSS is a vision-targeted instrument designed to measure the impact of ocular surface disease on vision-related quality of life in glaucoma patients. The questionnaire includes 10 ocular complaints, some of nonvisual nature (burning/smarting/stinging, tearing, dryness, itching, soreness/tiredness, feeling of something in the eye: “GSS Symptom score”) and some of visual nature (blurry/dim vision, hard to see in daylight, hard to see in dark places, halos around lights: “GSS Function score”), common among patients treated for glaucoma. For each eye, the questionnaire provides a score ranging from 0 to 100, with 0 representing the presence of a very bothersome problem and 100 representing the absence of a problem. The final GSS score is an unweighted average of the responses to all 10 items; the GSS subscale symptom score (Symp-6) is an unweighted average of all items that comprise the particular subscale.

Visual acuity was recorded in decimal notation unit, it was evaluated with the patients’ current lens correction (distance and near).

Slit lamp examination of the anterior and posterior segment was performed with particular attention to the ocular surface status evaluated with the tear film break-up time (tf-BUT) and corneal staining.

To measure the tf-BUT, a fluorescein-impregnated strip was placed in the inferior fornix and the patient was asked to blink several times; the average value of three measurements was calculated. tf-BUT values greater than or equal to 10 s were coded as normal and tf-BUT values less than 10 s as abnormal.

The presence of corneal staining was defined as more than one dot of fluorescein staining over the corneal surface.

## 3. Statistical Issues

### 3.1. Sample Size and Power Calculation

In the hypothesis of a target Cronbach’s alpha of 0.9 (as in the original Spanish version), a sample size of 150 subjects (120 VDT workers and 30 controls) allowed 95% confidence intervals between 0.88 and 0.92. This sample size also allowed the testing of Cronbach’s alpha within the following categories: gender, presbyopia (yes/no), VDT worker vs. control (for testing, in a 17-item questionnaire, a Cronbach’s alpha of 0.9 against 0.8 with a 90% power and alpha error of 5%, 50 subjects are required).

### 3.2. Statistical Analysis

Descriptive statistics were produced for the demographic, clinical and laboratory characteristics of cases. The groups were compared with parametric or nonparametric tests, according to the data distribution, for continuous variables, and with Pearson’s chi-squared test (Fisher’s exact test where appropriate) for categorical variables. Correlation between continuous variables was assessed by means of Pearson’s or Spearman’s coefficients, according to the data distribution. The Shapiro–Wilk’s test was used to test the normal distribution of quantitative variables. In all cases, 2-tailed tests were used. The *p*-value significance cutoff was 0.05.

The following analyses were performed:Scoring of the questionnaires according to the Spanish version rules (1–4).Evaluation of the completeness at the item, dimension and scale level.Calculation of the descriptive statistics at the dimension and scale level (mean and 95% CI, percentage of ceiling and floor) and at the group level.Reliability (Cronbach’s alpha) for internal consistency (primary objective).Multitrait/multi-item correlation matrix to assess:
The item’s internal consistency (item-scale correlation, corrected for overlap).The equality of item-scale correlations.The item’s discriminant validity (item-scale correlation vs. correlation with other scales and indicators: visual acuity, break-up time, corneal staining, GSS questionnaire).
Test–retest correlation for temporal stability (questionnaires completed twice 5–7 days apart): Pearson’s correlation coefficient and intraclass correlation coefficient.Factor analysis.Comparisons between subjects and controls of the 2 dimensions of the scale, for each eye, by means of the Mann–Whitney test, to verify discriminative property.External validity: Association of CVSS17 with:
The GSS questionnaire (Pearsons’s correlation coefficient)The number of years since working at a VDT, number of effective hours at a VDT per day, number of breaks (hours) per day, visual acuity (VA), tear film break-up time (tf-BUT), corneal staining (linear regression models).


A principal component analysis (PCA) was applied to confirm dimensionality.

An item response theory analysis was applied to assess the properties of the translated CVSS. Partial credit models (PCM) were employed [21].

For individual items the following analyses were made:Boundary characteristic curves (BCC) to determine the estimated category difficulties (the point at which a person with ability equal to a given difficulty has a 50% chance of responding in a category equal to or higher than the difficulty designates).Category characteristic curves (CCC) (probabilities of respondents choosing exactly category k; the points where the adjacent categories cross-represent transitions from one category to the next). To assess model fit, the empirical response proportions were plotted against the predicted latent trait (difficulty).To indicate the amount of information an item provides for estimating the latent trait, we plotted individual item information functions (IIF).

For the entire scale the following analyses were made:The test characteristic curve (TCC) (plot of the expected score against the latent trait) was calculated; the expected score at the theta value of ±1.96 was calculated.Moreover, we plotted the summated score versus the predicted score, to assess fit.Finally, we obtained a test information function (TIF) plot (test and its standard error), to indicate how well the instrument can estimate person locations.

Stata computer software version 14.0 (Stata Corporation, 4905 Lakeway Drive, College Station, TX 77845, USA) was used for the statistical analysis.

## 4. Results

### 4.1. Clinical Data

We enrolled 216 subjects: 128 VDT workers (41 men and 87 women) and 88 controls (38 men and 50 women) (Table 1). The mean age of the whole group was 42.7 ± 15.3 years. One patient signed informed consent and underwent all ophthalmic examinations but did not complete the CVSS17 questionnaire. The two groups were similar apart from the age (*p* = 0.002), the time spent at video terminals (*p* = 0.0001), the years since working at a VDT (*p* = 0.0001), the number of breaks and time spent on a VDT per day (all *p* < 0.001).

Regarding clinical data (Table 1), there was a statistically significant difference in tf-BUT (*p* = 0.002) and in corneal staining in both eyes between VDT workers and controls (*p* = 0.001 for right eye and 0.003 for left eye). Conversely, no difference between groups was found for the following parameters: best corrected visual acuity, intraocular pressure, retinal diseases and use of lacrimal substitutes. GSS questionnaire was statistically different between groups with significant better scores for controls (*p* < 0.0001).

The 17 CVSS items were completely answered by all subjects. The floor and ceiling effect for the questionnaire were within the predefined quality ranges.

### 4.2. Reliability

The overall Cronbach’s alpha was extremely high (0.925) and was always lower (from 0.917 to 0.924) (see Table 2) if individual items were removed. Items were well correlated with the entire test (item-test correlation) (Table 2); in addition, item-retest correlations were lower than item-test correlation, and not exceedingly higher than average interitem correlation (Table 2).

The CVSS17 test–retest reliability for temporal stability was high (0.91, *p* < 0.001) (Figure 1).

### 4.3. Validity

The control group of participants had better scores across all CVSS17 items (Table 3).

Association between CVSS17 and patients’ characteristics (Table 4) was found for age, gender, years worked at a VDT, years since starting using a VDT, time spent on breaks, BCVA, tf-BUT, corneal staining, use of lacrimal substitutes, presbyopia and GSS (both symptoms and function). Of note, the effect size of the association between the CVSS17 score with the GSS symptom scale (−0.287, *p* < 0.001) was higher than with the GSS function scale (−0.197, *p* < 0.001). The correlation coefficients between individual CVSS17 items and GSS symptom scale ranged from 0.38 to 0.78 (all *p* < 0.001) (data not shown).

### 4.4. Principal Component Analysis (PCA)

PCA analysis confirmed two main components factor 1 including CVSS2-3-4-5-6-7-10-11-12-13-14-16 and factor 2 including CVSS1-8-9-15-17; in the Spanish version, item 11 was in factor 2; the item contribution to the first component was similar (Figure 2a,b).

### 4.5. Rasch Analysis

#### 4.5.1. Analyses for Individual Items (BCC, CCC, IIF)

The boundary characteristic curves (BCC) plots confirmed that categories within items were correctly ordered in terms of “difficulty” for all items (Figure 3).

Regarding the category characteristic curves (CCC), the probability of choosing each increasing category was also higher for higher “difficulty” (Figure 4) for all items; there was good fit to the model for most items and response categories.

The item information function (IIF) was low for items 9-11-12-14 (Figure 5).

#### 4.5.2. Analyses for the Entire Scale (TCC, TIF)

A plot of the expected score against the latent trait revealed that patients with scores <17.9 and >43.6 represented less than 5% of the estimated distribution (Figure 6).

The graphical assessment of model fit to the observed summated score appeared satisfactory (Figure 7).

The CVSS can estimate with good precision the latent trait for most values; as expected, the precision was lower for subjects scoring very low or very high, as indicated by the test information plot with its standard errors (Figure 8).

A comparison of the Italian and Spanish versions outlined that some items showed a different ability to capture symptoms (Table 5).

## 5. Discussion

The perception of visual functioning by video terminal workers in relation to their health-related quality of life should be incorporated into ophthalmological clinical care. The recent COVID-19 pandemic has increased the number of subjects who spend many hours at a terminal and consequently the number of people around the world who complain of disturbances related to prolonged use of video terminals has increased.

The use of validated and reliable questionnaires is mandatory to evaluate the quality of life of these subjects; moreover, it represents a useful tool to follow VDT workers’ discomfort over time.

The CVSS17 is a questionnaire specifically designed to assess the quality of life of VDT workers.

Our study has shown that the Italian version of CVSS17 has psychometric properties comparable to those of the Spanish version; therefore, it represents a specific tool for measuring the quality of life in Italian subjects working at a VDT.

The results of our study suggest that the questionnaire has good validity, discriminatory power, internal consistency and reliability. Regarding the reliability, the high Cronbach’s alpha coefficients and the factor analysis indicate that the CVSS17 items are interdependent and homogeneous in terms of the concepts they measure. The original Spanish questionnaire showed comparable reliability (Cronbach’s alpha of 0.92) [6]. As indicated by the item-scale correlation analysis, all items appear to contribute to the quality of the scale.

The high temporal stability of the CVSS17 was demonstrated by the strength of the test–retest reliability and of the intraclass correlations: these are critical characteristics for a questionnaire to be used in long follow-up studies [16].

Concerning the external validity, a good correlation between the responses to the CVSS17 and analogous domains of the GSS was demonstrated, even if the two questionnaires are capturing different dimensions of health, as suggested by the finding that about half of the CVSS17 items showed a correlation coefficient lower than 0.5 with GSS.

Regarding the scores, the mean score of the questionnaire was similar for the Italian and the Spanish version, respectively, 31.3 ± 7.3 and 30.8 ± 7.5 [6]. The control group had better scores confirming a good discriminatory power between VDT workers and controls for the translated version.

The principal component analysis indicated two main factors (like in the Spanish version) that might be summarized as factor 1, dry eye symptoms and signs, and factor 2, visual impairment and photophobia. We noted that item 11, which belonged to factor 2 in the Spanish version, belonged to factor 1 in the Italian version: this item investigates photophobia and is similar to item 17, which belonged to factor 2; photophobia is also a symptom of dry eye.

The Rasch analysis of the Italian CVSS had a good fit to the data; it is therefore reliable in its conclusions, indicating a fair performance of the response categories within items. Regarding the entire scale, values <18 probably indicated the absence of ocular problems and values >44 might deserve a more careful ophthalmological evaluation of the ocular surface conditions. The Italian CVSS17 can be considered precise in its assessment of ocular surface symptoms of discomfort. The low information provided by some items indicated that some items might be redundant in the Italian version: item 9 was also covered by item 8, item 11 by item 17, item 12 by item 6, and item 14 by item 4.

## 6. Conclusions

In conclusion, the results of the CVSS17 validation process in Italian support the validity and reliability of the questionnaire and the overall score represents a good measure of the symptoms of the patients working at a VDT. The test can be applied to measure the vision-related quality of life in VDT users in an Italian setting both for clinical and for research purposes. It is likely that a new, shorter version might be built for the Italian setting; however, this would impair the crosscultural comparison.

Patients’ well-being has become crucial in all the most recent studies regarding ocular diseases. The prolonged use of a VDT can cause and/or induce the onset of visual, ocular and even systemic disorders that can impact the patients’ quality of life: the validated Italian version of the CVSS17 will facilitate the clinicians’ work in detecting the impact of visual concerns in these subsets of subjects.

## Figures and Tables

**Figure 1 ijerph-19-02517-f001:**
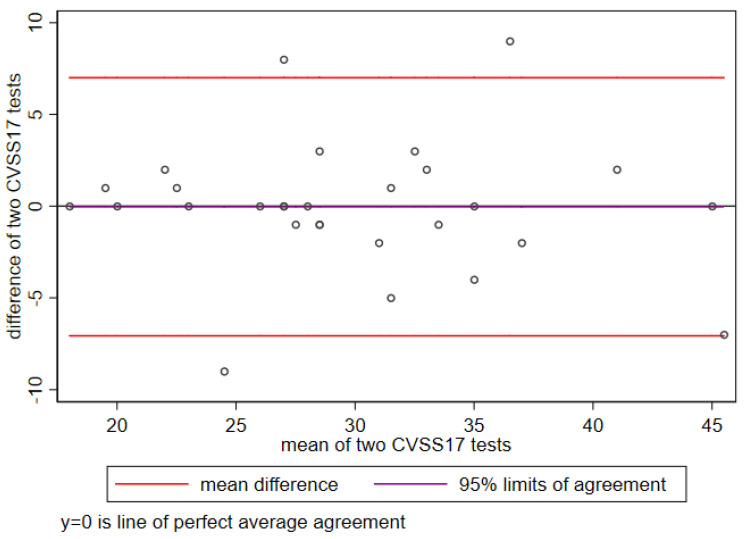
Reliability: test–retest for temporal stability of CVSS17 in a random subset of 20 subjects.

**Figure 2 ijerph-19-02517-f002:**
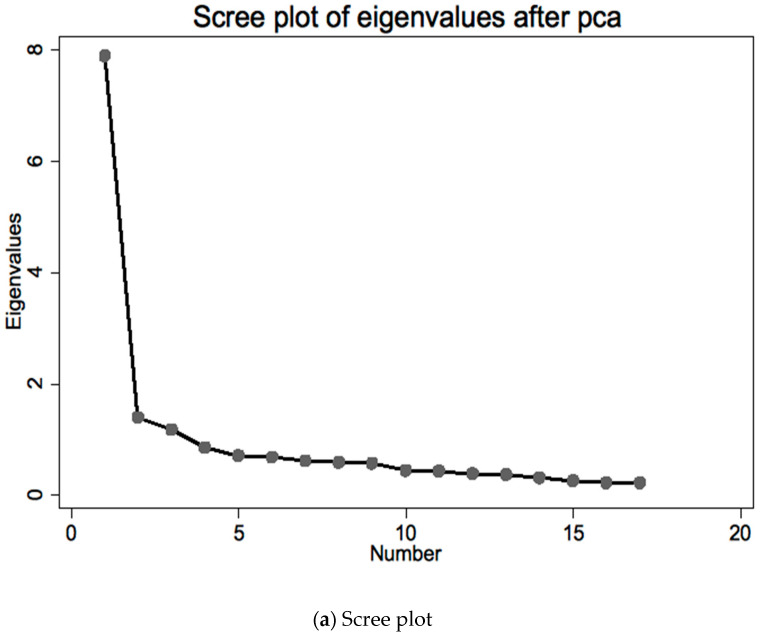
Principal components analysis of the Italian CVSS: scree plot and loadings plot (rotated varimax). (**a**) Explanation: this is a useful way to decide how many dimensions are represented by the data; it plots the dimensions as the *X*-axis and the corresponding eigenvalues (variance) as the *Y*-axis (the dimension with the largest eigenvalue has the most variance, dimensions with smaller or negative eigenvalues are negligible; traditionally, only eigenvalues >1 are considered relevant). Across subsequent dimensions, eigenvalues decline; the number of dimensions necessary to explain the data is indicated by the number of dimensions before the “elbow” (the point where the slope of the curve flattens out). Comment: in our questionnaire, the computer vision scale is well represented by a single dimension. (**b**) Explanation: it shows the loadings (correlations of the items with the dimension) for the two dimensions with highest variance in each area of awareness. Comment: most items are similarly correlated to the first dimension, and unevenly correlated with the second dimension retained in the analysis; these plots, besides confirming the findings from the multitrait/multi-item analysis, support the correct dimensionality of the questionnaire.

**Figure 3 ijerph-19-02517-f003:**
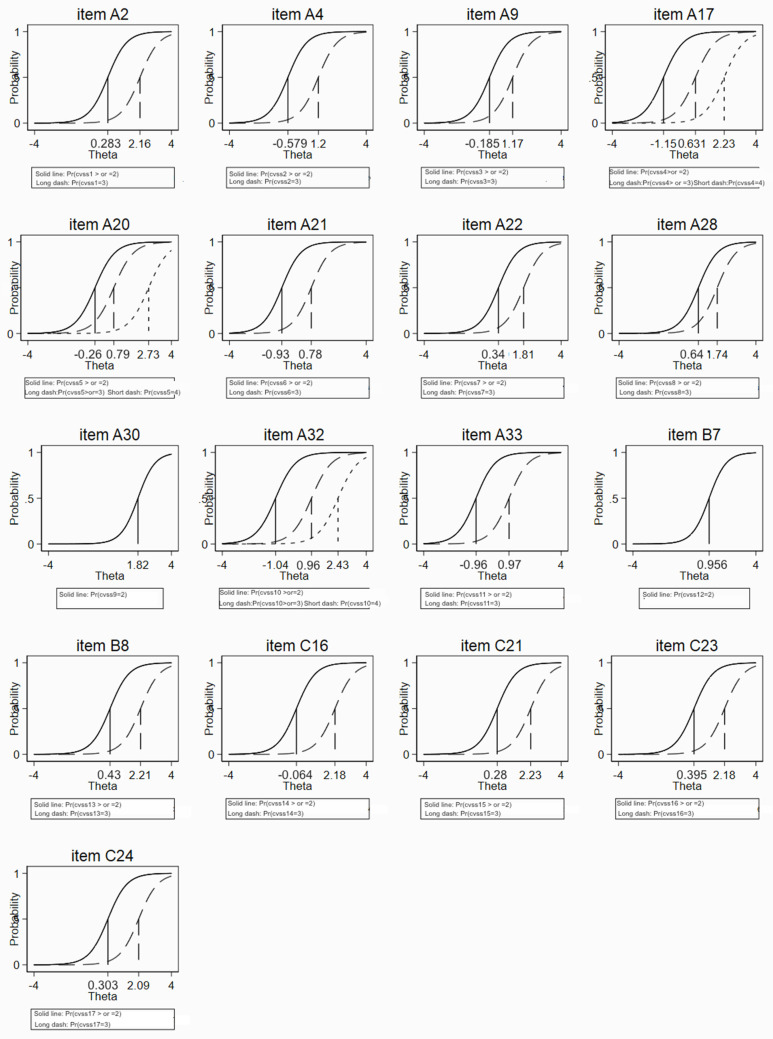
Boundary characteristic curves (BCC) from the graded response model (IRT analysis) of the Italian version of CVSS.

**Figure 4 ijerph-19-02517-f004:**
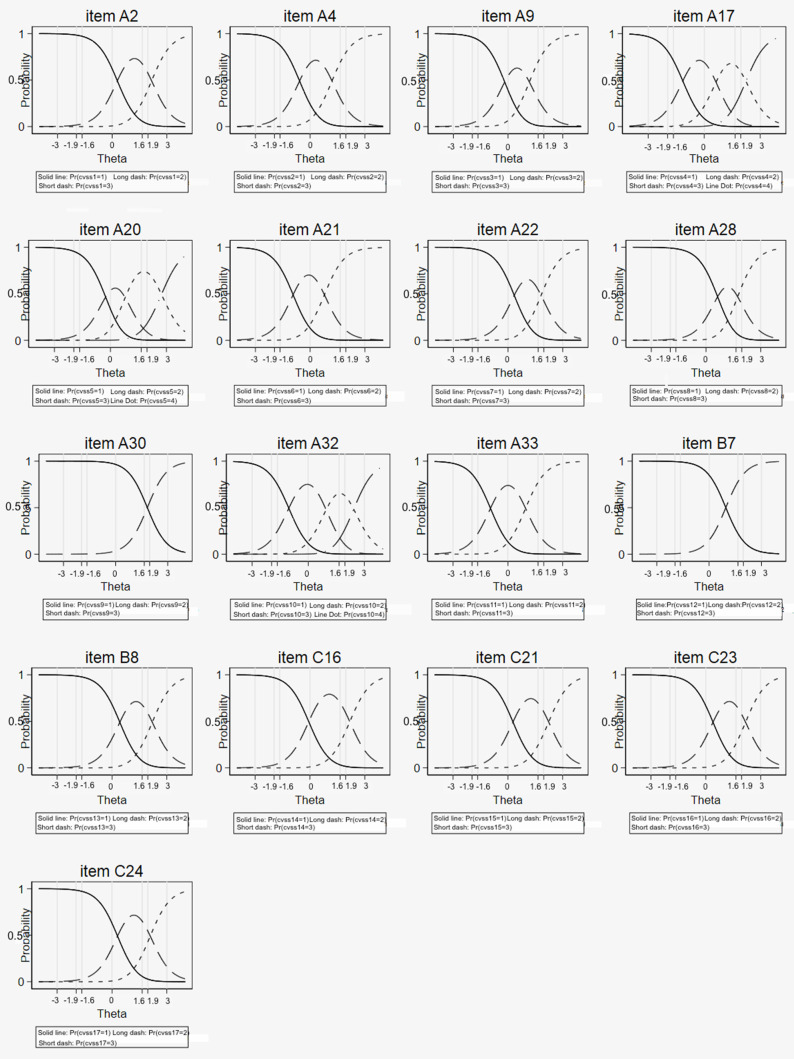
Category characteristic curves (CCC) and empirical proportions for each of the 17 CVSS items from the graded response model (IRT analysis) of the Italian version of the CVSS17.

**Figure 5 ijerph-19-02517-f005:**
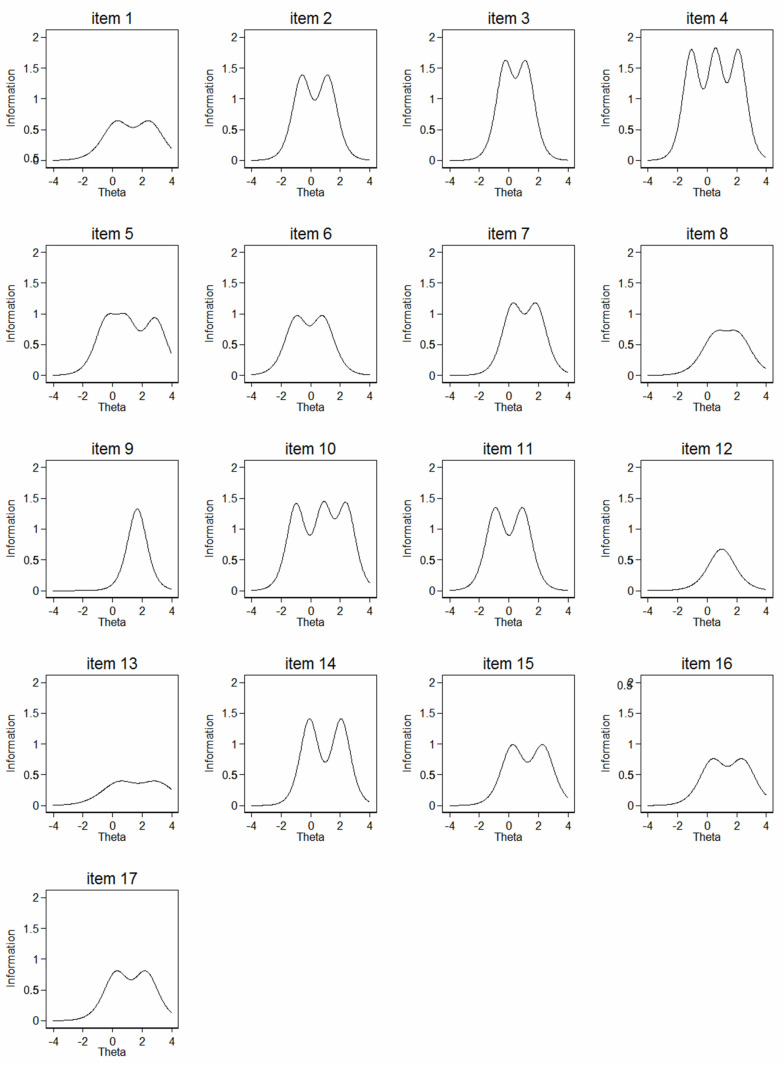
Item information function (IIF) proportions for each of the 17 CVSS items from the graded response model (IRT analysis) of the Italian version of CVSS.

**Figure 6 ijerph-19-02517-f006:**
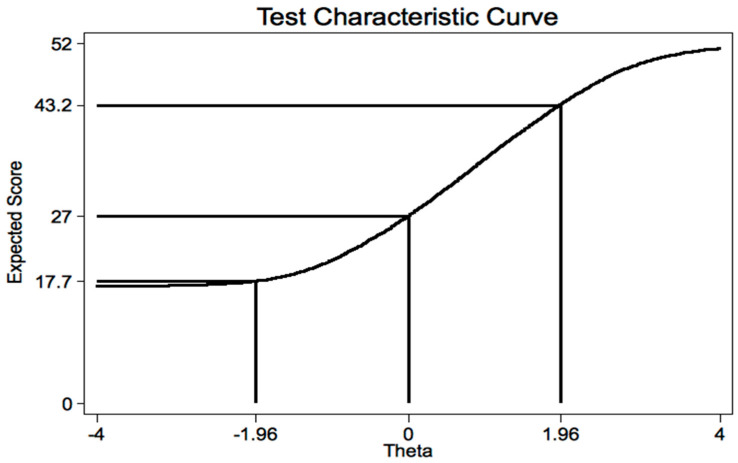
Test characteristic curve (TCC) from the partial credit model (IRT analysis) of the Italian version of CVSS.

**Figure 7 ijerph-19-02517-f007:**
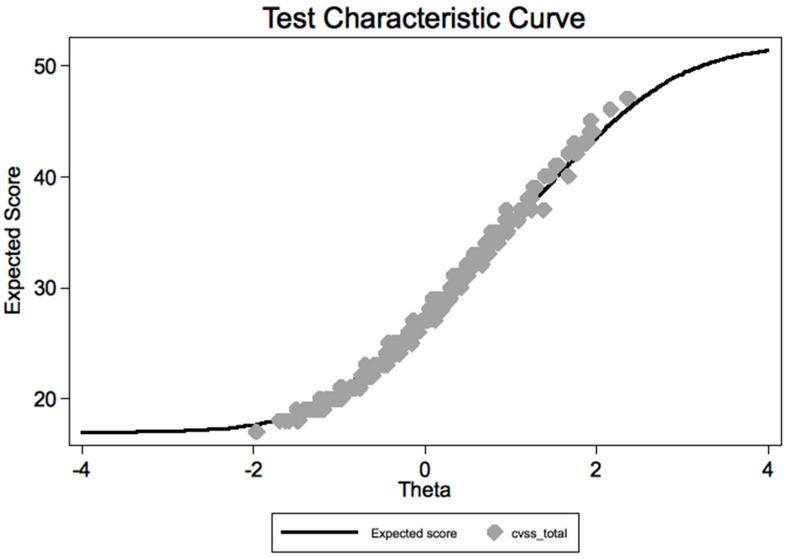
Test characteristic curve (TCC): CVSS summated score vs. predicted score from the graded response model (IRT analysis) of the Italian version of the CVSS.

**Figure 8 ijerph-19-02517-f008:**
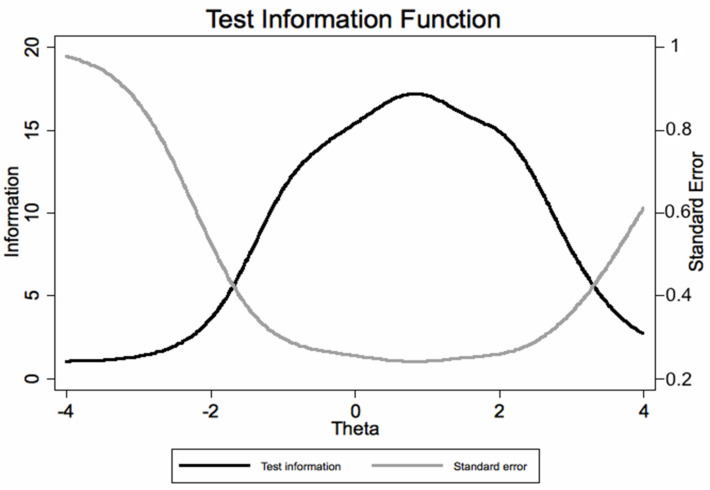
Test information function (TIF) from the graded response model (IRT analysis) of the Italian version of the CVSS.

**Table 1 ijerph-19-02517-t001:** Patients’ sociodemographic and clinical characteristics (VDT = video terminal workers; BCVA = best corrected visual acuity; IOP = intraocular pressure; tf-BUT = tear film break-up time; R = right eye; L = left eye; GSS = Glaucoma Symptom Scale; Y = yes; ys = years; h = hours; N = number; bold font: statistically significant *p*-value).

Variable			Controls		VDT	*p*-Value
Number			88		128	
Women/men		N	50/38		87/41	0.114
Schooling level	8 Years	N	6		11	
	13 Years	N	41		57	
	>13 Years	N	41		60	0.631
IRCCS San Matteo Foundation employees		N (%)	43 (48.8)		83 (64.9)	0.025
Systemic comorbidities (Y)		N (%)	19 (21.6)		33 (25.8)	0.52
Presbyopia (Y)		N (%)	14 (46.7)		48 (42.6)	0.662
Age (ys)		Median (IQR)	32.4 (26.4–53.2)		46.7 (34.3–53.7)	**0.002**
Years since VDT users (ys)		Median (IQR)	0 (0–0)		10 (5–20)	**0.0001**
Hours spent at VDT/day (h)		Median (IQR)	2 (1–3)		7.2 (6–8)	**0.0001**
Breaks/day (N)		Median (IQR)	0 (0–1)		2 (1–3)	**0.0001**
Breaks/day (h)		Median (IQR)	00:00 (00:00–00.30)		01:00 (00:45–01:00)	**0.0001**
BCVA (decimals)		Median (IQR)	R	10 (10–10)	10 (10–10)	0.39
			L	10 (10–10)	10 (10–10)	0.663
IOP (mmHg)		Median (IQR)	R	14 (14–14)	14 (14–15)	0.064
			L	14 (14–15)	14.5 (14–15)	0.086
tf–BUT (seconds)		Median (IQR)	R	10 (9–12)	9 (7–10)	**0.0002**
			L	10 (9–12)	9 (7–10)	**0.0002**
Corneal staining (Y)		N (%)	R	2 (4.2)	30 (25.9)	**0.001**
			L	2 (4.2)	27 (23.3)	**0.003**
Retinal fundus (diseases) (Y)		N (%)	R	1 (1.1)	7 (7.9)	0.161
			L	0 (0.0)	5 (3.9)	0.372
Lacrimal substitute use (Y)		N (%)	R	8 (27.6)	19 (17.9)	0.295
			L	8 (27.6)	19 (17.9)	0.295
GSS total		Median (IQR)	95 (85–100)		77.5 (67.5–85)	**0.0001**
GSS symptom		Median (IQR)	91.7 (79.2–100)		75 (58.3–83.3)	**0.0001**
GSS function		Median (IQR)	100 (100–100)		87.5 (71.9–100)	**0.0001**

**Table 2 ijerph-19-02517-t002:** Reliability: item-scale correlations and Cronbach’s alpha of the Italian version of the CVSS17.

Item	Item-Test Correlation	Item-Retest Correlation	Average Interitem correlation (IC)	Cronbach’s Alpha
A2	0.5923	0.5310	0.4266	0.9225
A4	0.7360	0.6918	0.4129	0.9184
A9	0.7669	0.7271	0.4100	0.9175
A17	0.7595	0.7186	0.4107	0.9177
A20	0.7245	0.6788	0.4140	0.9187
A21	0.7384	0.6945	0.4127	0.9183
A22	0.7002	0.6514	0.4163	0.9194
A28	0.6224	0.5644	0.4237	0.9217
A30	0.5390	0.4726	0.4317	0.9240
A32	0.7709	0.7316	0.4096	0.9173
A33	0.7408	0.6973	0.4124	0.9182
B7	0.6310	0.5738	0.4229	0.9214
B8	0.5692	0.5056	0.4288	0.9231
C16	0.6711	0.6187	0.4191	0.9203
C21	0.6714	0.6189	0.4191	0.9203
C23	0.5842	0.5221	0.4274	0.9227
C24	0.6250	0.5672	0.4235	0.9216
Whole scale			0.4189	0.9246

**Table 3 ijerph-19-02517-t003:** Validity of the CVSS17 items: *p*-valuecomparison of the item scores between VDT workers and not (bold font: statistically significant).

Item	VDT Workers Mean ± SD	No VDT Workers Mean ± SD	*p*-Value
A2	1.71 ± 0.63	1.13 ± 0.34	**0.0001**
A4	2.16 ± 0.66	1.42 ± 0.56	**0.0001**
A9	2.03 ± 0.74	1.45 ± 0.62	**0.0001**
A17	2.50 ± 0.75	1.58 ± 0.67	**0.0001**
A20	2.09 ± 0.87	1.55 ± 0.77	**0.002**
A21	2.17 ± 0.75	1.74 ± 0.73	**0.005**
A22	1.73 ± 0.69	1.29 ± 0.53	**0.001**
A28	1.59 ± 0.71	1.13 ± 0.34	**0.0006**
A30	1.17 ± 0.38	1.03 ± 0.18	**0.050**
A32	2.21 ± 0.78	1.74 ± 0.82	**0.005**
A33	2.23 ± 0.67	1.48 ± 0.63	**0.0001**
B7	1.33 ± 0.47	1.03 ± 0.18	**0.0008**
B8	1.54 ± 0.64	1.32 ± 0.54	**0.051**
C16	1.82 ± 0.58	1.42 ± 0.50	**0.0007**
C21	1.69 ± 0.59	1.26 ± 0.58	**0.0001**
C23	1.64 ± 0.64	1.26 ± 0.44	**0.002**
C24	1.69 ± 0.64	1.26 ± 0.51	**0.0004**
CVSS tot	31.30 ± 7.37	23.10 ± 5.44	**0.0001**

**Table 4 ijerph-19-02517-t004:** Validity of the CVSS17: association between CVSS17 score and patients’ characteristics (univariable linear regression models) (F = female; M = male; VDT = video terminal workers; N = number; h = hours; Δ = diopters; tf-BUT = tear film break-up time; IOP = intraocular pressure; GSS = Glaucoma Symptom Scale; (bold font: statistically significant *p*-Value).

Variable	Category/Description	Regression Coefficient	95% Confidence Intervals	*p*-Value
Age		0.112	0.012 to 0.211	**0.029**
Gender	F vs. M	−4.033	−6.51 to −1.56	**0.001**
Schooling level	8–13 years vs. <8	0.199	−4.07 to 4.47	0.927
	>13 vs. <8 years	1.339	−2.93 to 5.60	0.538
Systemic comorbidities	Yes vs. No	4.643	1.71 to 7.57	**0.002**
IRCCS employees	No vs. Yes	1.148	−1.63 to 3.93	0.418
Years since VDT users		0.216	0.088 to 0.344	**0.001**
Hours spent at VDT/day		1.345	0.953 to 1.74	<**0.001**
Breaks/day (N)		0.119	−0.696 to 0.936	0.773
Breaks/day (h)		<0.001	0 to 0	**0.03**
BCVA (decimals)		−1.685	−2.37 to −1.01	<**0.001**
Optical correction (distance)	Yes vs. No	0.733	−1.77 to 3.24	0.567
Optical correction (near)	Yes vs. No	2.135	−0.43 to 4.71	0.104
Presbyopia	Yes vs. No	2.889	0.33 to 5.45	0.027
Fusional Vergence (Δ)(near)		−0.07	−0.235 to 0.096	0.411
Fusional Vergence (Δ) (distance)		−0.116	−0.289 to 0.058	0.193
IOP (mmHg)		1.044	−0.037 to 2.125	0.058
tf–BUT (seconds)		−1.238	−1.614 to −0.863	<**0.001**
Corneal staining (grade)	Yes vs. No	6.598	3.73 to 9.465	<0.001
Lacrimal substitute use	Yes vs. No	7.639	4.72 to 10.56	<0.001
Retinal fundus (pathologies)	Yes vs. No	3.038	−1.77 to 7.85	0.216
GSS total		−0.374	−0.42 to −0.329	<**0.001**
GSS symptom		−0.287	−0.319 to −0.254	<**0.001**
GSS function		−0.197	−0.254 to −0.14	<**0.001**

**Table 5 ijerph-19-02517-t005:** Comparison of Italian and Spanish version of the CVSS for all item measures.

Item	Spanish Version		Italian Version	
	Measure	Error	Measure	Error
A2	0.97	0.09	0.65	0.16
A4	−1.49	0.08	−1.34	0.14
A9	0.12	0.08	−1.04	0.14
A17	−0.92	0.08	−0.98	0.13
A20	0.17	0.07	0.05	0.13
A21	−0.71	0.08	−0.29	0.13
A22	−0.11	0.08	1.50	0.15
A28	1.39	0.09	0.33	0.15
A30	0.99	0.12	1.60	0.28
A32	0.27	0.08	−0.50	0.13
A33	−0.95	0.08	−2.02	0.14
B7	−0.34	0.10	−0.16	0.20
B8	0.38	0.08	0.51	0.16
C16	−0.44	0.08	−0.04	0.15
C21	−0.07	0.09	0.70	0.16
C23	−0.09	0.08	0.65	0.16
C24	0.83	0.09	0.38	0.16

## Data Availability

Data are available from Authors.

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
