# Peer review of "The 17-Item Computer Vision Symptom Scale Questionnaire (CVSS17): Translation, Validation and Reliability of the Italian Version"

_ijerph, 2022, doi:10.3390/ijerph19052517_

Round 1

Reviewer 1 Report

The manuscript is  a solid work, describing the study that authors conducted to validate the questionnaire. It will be very helpful if they attach the questionnaire itself as a supplement or appendix. I tried to find it but only saw the study from Spain which also did not present the questionnaire. 

Author Response

The manuscript is a solid work, describing the study that authors conducted to validate the questionnaire. It will be very helpful if they attach the questionnaire itself as a supplement or appendix. I tried to find it but only saw the study from Spain which also did not present the questionnaire.

A. Thank you very much for your comments and for your suggestion: I've attached the questionnaire.

Reviewer 2 Report

This manuscript is necessary for its correct use in Italy. However, the manuscript has to be improved for acceptance:

Methodology:
- By whom was the questionnaire translated? Has the reverse and direct translation been performed?

Results:
In general, the results section is very difficult to understand. I recommend rearranging everything.
- How was the sample size obtained?
- Line 89, write +/- sign
- Divide the subsections into 4.1,4.2….
- The figure, they don't understand each other. Please explain them in more detail
- Line 218, put (p <0.001)
- Put the figures together with the text

Author Response

This manuscript is necessary for its correct use in Italy. However, the manuscript has to be improved for acceptance:

Methodology:

- By whom was the questionnaire translated? Has the reverse and direct translation been performed?

A. thank you for your observation, I've revised and better explained this process in Methods (lines 94-109).

Results:

In general, the results section is very difficult to understand. I recommend rearranging everything.

- How was the sample size obtained?

A. The sample size is described on lines 154-159. Since the sampling of subjects occurred in several wards, more health care workers than expected volunteered to participate. Rather than excluding them, we retained a larger sample size than necessary to test the original hypothesis, and to achieve greater precision. That is why the results refer to 216 subjects.

- Line 89, write +/- sign

A. Done

- Divide the subsections into 4.1,4.2….

A. Thanks for the suggestion: I added the subsections (lines 257-396), now the results are clearer and more understandable

- The figure, they don't understand each other. Please explain them in more detail

A. thanks for this comment: I added explanation at each figure.

- Line 218, put (p <0.001)

A. Thank you for the suggestion, I did it (now line 224).

- Put the figures together with the text

A. Done

Reviewer 3 Report

Journal: IJERPH (ISSN 1660-4601)

Manuscript ID: ijerph-1491856

Type: Article

Title: The 17-item Computer Vision Symptom Scale questionnaire (CVSS17): Translation, Validation and Reliability of the Italian version.

Comments: Major revision

Thank you for the opportunity of reviewing this manuscript (ijerph-1491856). I would like to congratulate the authors for this study on a very important issue. In my opinion this is a piece of research of academic and clinical interest. I found the manuscript interesting and with an important impact on a key measurement of Computer Vision Symptom (CVSS17). However, I have some questions about some crucial points:

  • The translation process on the CVSS17 is reported in very little detail, which makes it both not very transparent nor is there much to learn for readers about the obstacles encountered in this translation between the Spanish and Italian languages. Pls describe in details follow one of guidelines, such as on the WHO.

  • IRT application requires two important assumptions: (1) the construct being measured is in fact unidimensional and (2) the items display local independence. Pls further clarify these two points.

  • Pls elaborate on the distribution of the item data? normality or any mild-to-moderate violations of normality, even…?

  • Pls report psychometric properties. according to guidelines on the COMIN (my major concern).

  • More information about the sampling method is needed here (3.1. Sample size, P3); furthermore, where, when, and how were participants approached? What mode of survey administration was used (online or paper-and-pencil)?

  • 2. Statistical analysis, P3: It is unclear whether the items were treated as ordinal in this paper (my major concern). Pls clarify the estimation method used. The authors might apply to Weighted Least Square Mean and Variance adjusted (WLSMV)/Robust Diagonally Weighted Least Squares (RDWLS) to accommodate categorical data.

  • Line 189, P4: 42.7?15.3 years; Table 5, P5: pls add note for font bold… Pls check the full text for similarities.

  • Pls update some limitations of this study. The manuscript can benefit from improvement of English by a professional Anglophone editor.

Kind regards,

Your reviewer

Author Response

Thank you for the opportunity of reviewing this manuscript (ijerph-1491856). I would like to congratulate the authors for this study on a very important issue. In my opinion this is a piece of research of academic and clinical interest. I found the manuscript interesting and with an important impact on a key measurement of Computer Vision Symptom (CVSS17). However, I have some questions about some crucial points:

The translation process on the CVSS17 is reported in very little detail, which makes it both not very transparent nor is there much to learn for readers about the obstacles encountered in this translation between the Spanish and Italian languages. Pls describe in details follow one of guidelines, such as on the WHO.

A. thank you for your comment, I revised and rewrote the translation process with more details (lines 94-109).

The steps for getting the Italian adaptation process were: translation into Italian, assessment of item comprehension, back-translation into English, development of a consensual version, and formal assessment of its validity and reliability [15]. Two bilingual translators (GLP and GCR) translated the questionnaire into Italian (GCMR and GMP), rating difficulty from 0 (lowest) to 100 (highest). Conceptual rather than literal equivalence aimed at. Discrepancies were reconciled. Another translator (LS) rated the reconciled forward translation for conceptual equivalence to the Spanish version (scale 1-100). The translated questionnaire has been read to 10 Italian subjects, to verify clarity, un-ambiguity and completeness of each item. A bilingual Spanish-speaking translator (Isabel Mazzucchelli) translated back the questionnaire into English and then rated the equivalence to the original Spanish version (1-100). The Italian version was then discussed by a panel of experts to obtain content correspondence. We consulted the method for cross-cultural equivalence in psychiatric research described by Ware for the IQOLA project in 1998 [16] and McAlinden in 2011 [17]".

IRT application requires two important assumptions: (1) the construct being measured is in fact unidimensional and (2) the items display local independence. Pls further clarify these two points.”

A. the statistician had checked everything, according to the literature, at the time of validation, unfortunately now I no longer have her available, since she no longer works in Pavia.

If you prefer I can delete the IRT analysis, but this analysis was performed correctly, following the rules and the assumptions that underlie it.

Pls elaborate on the distribution of the item data? normality or any mild-to-moderate violations of normality, even…?

A: The Shapiro-Wilk’s test was used to test the normal distribution of quantitative variables. The test was added in “Statistical Analysis” (line 166)

Pls report psychometric properties. according to guidelines on the COMIN (my major concern).

A. The psychometric properties have been reported according to the bibliographical indications I have cited: the main and most important properties have been analyzed and described. We used the method for cross-cultural equivalence in psychiatric research described by Ware for the IQOLA project in 1998 [Ware JE, Gandek B. Methods for testing data quality, scaling assumption, and reliability: the IQOLA project approach. International Quality of Life Assessment. J Clin Epidemiol 1998;51:945-52] and McAlinden in 2011 [McAlinden C, Khadka J, Pesudovs K. Statistical methods for conducting agreement (comparison of clinical tests) and precision (repeatability or reproducibility) studies in optometry and ophthalmology. Ophthalmic and Physiological Optics 2011; 31: 330-338]

More information about the sampling method is needed here (3.1. Sample size, P3);

A. The sample size is described on lines 154-159. Since the sampling of subjects occurred in several wards, more health care workers than expected volunteered to participate. Rather than excluding them, we retained a larger sample size than necessary to test the original hypothesis, and to achieve greater precision. That is why the results refer to 216 subjects.

furthermore, where, when, and how were participants approached?

A. I've better explained , lines 113-116.

What mode of survey administration was used (online or paper-and-pencil)?

A. I've added the mode of administration, that was paper and pencil. (line 122).

Statistical analysis, P3: It is unclear whether the items were treated as ordinal in this paper (my major concern). Pls clarify the estimation method used. The authors might apply to Weighted Least Square Mean and Variance adjusted (WLSMV)/Robust Diagonally Weighted Least Squares (RDWLS) to accommodate categorical data.

A. Data were ordinal.

Line 189, P4: 42.7?15.3 years; Table 5, P5: pls add note for font bold… Pls check the full text for similarities.

A. editorial problems, I'll contact the editorial office

The manuscript can benefit from improvement of English by a professional Anglophone editor.

A. the paper has been revised by an Anglophone editor.

Reviewer 4 Report

Comments as per attached.

Author Response

line 21 and 26: thanks for the corrections.

Line 41: I've changed the sentence, thank you for suggestion.

Line 45: I agree, i've deleted the sentence.

Line 56: Thank you for the suggestion, I've deleted the sentence.

line 57: I've corrected. Thanks.

Line 58 but it is not a new instrument...you are basically making it valid for use in the Italian population?

A. lines 58-60 . thank you for this precisation: I've deleted “new” and changed the sentence accordingly.

Line 71. done.

Line 87: rephrase - did you mean "not VDT workers"?.....also subjects should not be referred to as patients here

A. Yes, I've changed it, thanks

lines 88-91:

A. The paraghaph has been re-written as follows: The employees of the San Matteo Foundation who operate as VDT workers and who carried out an eye examination for occupational medicine, were invited to participate in the study for the VDT group; patients who underwent an outpatient eye examination and who were not VDT workers were invited to participate in the study for the control group

line 92 Date of birth was collected, but age was used, I've changed as you suggested.

Line 93: Yes, thank you for the correction

line 100. with this then not have been affected by visit to the ophthalmologist?

A. Thank you for your observation, 5-7 days after the first visit , these 10 subjects completed only the questionnaire, they did not perform any eye examination.

Line 101, what was the use of this questionnaire to your study?

A. Thank you for this comment, I've added a sentence, in this way the use of GSS is better understood.

In order to provide data for the planned tests of correlational validity, the Italian version of the Glaucoma Symptom Scale (GSS) was administered in addition to the CVSS17. The GSS was chosen because it had been adequately translated and validated in Italian in 2013”.

line 114. thanks for all your suggestions. I've changed accordingly.

Lines 119-121.is this detailed description of the TBUT procedure necessary, as other procedures have not been detailed?

A. I've revised the description of BUT procedure. Thank you for your comment.

Line 128 and 131

A. Done

Line 128, but what was the sample size for your study, and how did you sample the subjects

A. The sample size is described on lines 154-159. Since the sampling of subjects occurred in several wards, more health care workers than expected volunteered to participate. Rather than excluding them, we retained a larger sample size than necessary to test the original hypothesis, and to achieve greater precision. That is why the results refer to 216 subjects.

Lines 164-176,needs to be properly aligned and formatted; line 190; line 218; line 273

A. editorial problem.

line 191. what was the mean age of each group? was one found to be more presbyopic than the other? From Table 1, control group non-presbyopic while VDT group presbyopic - this will have implications on VDT comfort....they needed to be matched....

A. the mean age was 37.8+/-14.9 for controls and age 44.8 +/-11.2 for VDT , madians are reported in table I; regarding presbyopia the two groups were matched: 46.7% of controls were presbyopes versus 42.6% of VDT workers, p=0.662.

line 373, why would this possibly change? Relevance?

A. thank you for this observation, I've delated the sentence

lines 413-16, have these two been cited in the manuscript?

A. Yes , the # 20 on line 130, the # 21 on line 193.

Round 2

Reviewer 2 Report

Accept in the current form

Author Response

Thank you for your comment

Reviewer 3 Report

Dear Authors,

Thank you for your careful revision of the manuscript. The changes have contributed greatly to improved understanding and clarity. However, the authors have only accepted some of my recommendations. There is a question about ordinal data form “Authors' Responses to Reviewer's Comments”. Line 182, Factor analysis: Please use Weighted Least Square Mean and Variance adjusted (WLSMV)/Robust Diagonally Weighted Least Squares (RDWLS) to accommodate categorical data using R or Mplus. EFA not equal to 4.4 Principal component analysis (PCA). Line 173, Reliability: Again, for reliability, additional value(s) for ordinal alpha is necessary. preferably using rank correlation, and so on.

Please don’t cut corners. Thank you for your work.

Best wishes,

your reviewer.

Author Response

Dear reviewer

Dr. scudeller no longer works with me in pavia, I consulted a second statistician, who told me that the analysis you requested is an analysis similar to the one we did but uses a different statistical program. therefore, in our opinion, the analyzes carried out and reported in the manuscript are sufficient to guarantee the scientific correctness of the validation of the questionnaire.